# Dietary Tryptophan Supplementation Implications on Performance, Plasma Metabolites, and Amino Acid Catabolism Enzymes in Meagre (*Argyrosomus regius*)

**Cláudia Teixeira** [1,2,*], **Rita Pedrosa** [1], **Carolina Castro** [3], **Rui Magalhães** [1,2], **Elisabete Matos** [4], **Aires Oliva-Teles** [1,2], **Helena Peres** [1,2] and **Amalia Pérez-Jiménez** [2,5,*]

1. Departamento de Biologia, Faculdade de Ciências, Universidade do Porto, Rua do Campo Alegre, Edifício FC4, 4169-007 Porto, Portugal
2. CIIMAR—Centro Interdisciplinar de Investigação Marinha e Ambiental, Universidade do Porto, Terminal de Cruzeiros do Porto de Leixões, Av. General Norton de Matos, 4450-208 Matosinhos, Portugal
3. FLATLANTIC–Atividades Piscícolas, S.A., Rua do Aceiro s/n, 3070-732 Praia de Mira, Portugal
4. Egas Moniz Center for Interdisciplinary Research, Egas Moniz School of Health and Science, 2829-511 Caparica, Portugal
5. Department of Zoology, University of Granada, Campus Fuentenueva s/n, 18071 Granada, Spain
* Correspondence: claudia.teixeira@ciimar.up.pt (C.T.); calaya@ugr.es (A.P.-J.)

**Abstract:** Tryptophan (Trp) is an essential amino acid, commercially available as a feed-grade product, and is a precursor to serotonin and melatonin, which are both important molecules in stress mitigation. Meagre have a high potential for marine aquaculture diversification but are highly susceptible to stressful conditions. This study aimed to assess the potential role of dietary tryptophan supplementation in meagre juveniles in order to minimize the deleterious effect of potential stress conditions. For this, a growth trial was performed wherein meagre juveniles were fed four isoproteic (45%DM) and isolipidic (16%DM) diets; namely, a control diet, and three diets similar to the control diet but supplemented with varying levels of tryptophan, graded according to the resulting percentage in each diet's dry matter (Control, 0.25%Trp, 0.5%Trp, and 1%Trp), corresponding to a total dietary tryptophan of 1.06, 1.70, 2.08, and 3.24 g 16 g$^{-1}$ N, respectively. Diets were tested in triplicate, and fish were fed twice a day, six days a week, for eight weeks. Five days after the end of the growth trial, a time-course blood sampling was performed at 0 h, 1 h, 3 h, 6 h, and 12 h after feeding. At the 6 h sampling point, the liver was also collected. Overall, our results indicate that 1Trp supplementation (total dietary Trp 3.24 g 16 g$^{-1}$ N) may be harmful to fish, decreasing growth performance and feed utilization, although doses up to 0.5Trp do not influence these parameters. Voluntary feed intake lineally decreased with the increase in the level of dietary Trp. Whole-body lipid content decreased at the highest tryptophan inclusion, whereas no changes were observed in protein levels. After 12 h from feeding, plasma glucose levels were lower in all dietary treatments supplemented with tryptophan compared to those observed in the control. Hepatic enzyme activity of protein catabolism decreased with dietary Trp inclusion. Overall, our results suggested that while a dietary Trp level increase of up to 2.08 g 16 g$^{-1}$ N did not affect growth performance and feed efficiency, both these parameters were severely compromised with a Trp level of 3.24 g 16 g$^{-1}$ N.

**Keywords:** amino acids; aquaculture; meagre; supplemented diets; tryptophan; welfare

## 1. Introduction

Interest in fish welfare is a matter of increasing importance in aquaculture. With the intensification of aquaculture practices, fish are submitted to daily stressful situations, including loss of water quality, handling, and transport, any of which may decrease growth performance due to the energy use toward stress mitigation, as well as increase their predisposition to diseases [1–3]. Stress is the reaction of animals to a threat, altering their

homeostasis, and consequently launching an endocrine stress response that releases cortisol into the bloodstream. To minimize stress, several options are being explored, including the use of specific nutrients or functional ingredients, such as probiotics, prebiotics, vitamins, minerals, nucleotides, and functional amino acids [4–6]. Tryptophan (Trp) is an essential amino acid required for protein synthesis and is the only precursor of the neurotransmitter serotonin (5-HT), which itself is the precursor for melatonin, both having a vital role in stress mitigation. In addition, it is commercially available as a feed-grade product, making the supplementation of Trp a viable and economic way to optimize both production and animal welfare [7]. Tryptophan may act as a functional amino acid due to its pivotal role in serotonin and melatonin synthesis, which are important in stress mitigation [8,9]. Previous studies reported that Trp-supplemented feeds affected fish behavior through 5-HT signaling, decreasing aggression, and attenuating stress-induced anorexia in several teleost species [8,10–13]. Moreover, studies have shown that Trp is an important limiting factor in serotonin synthesis, causing a dose-dependent response. A deficiency in Trp can result in affective disorders, anxiety, aggression, stress, and eating disorders, among other conditions [14–16]. Tryptophan-supplemented diets have been shown to lower plasma cortisol levels in *Oncorhynchus mykiss* [17]. Tryptophan is a widely studied functional amino acid. In meagre, the effects of a dietary supplementation of moderate levels (0.16 and 0.11% diet) have been studied in fish maintained under chronic or crowding stress [3,18]. Few reports have documented the effects of Trp-enriched diets on meagre, and the ones that have done so mainly tested on acute and chronic stressors.

Meagre (*Argyrosomus regius*) is a Mediterranean carnivorous species with a high potential for marine aquaculture diversification [19–23]. Meagre has several attractive qualities, such as a high growth rate and feed conversion efficiency, good nutritional value, and can grow in captivity, all of which are promising qualities for a future mass market species. Meagre feed requirements are similar to those of other Mediterranean marine species, such as gilthead seabream [19,20]. In 2020, European production of meagre reached almost 45,000 tones, representing an increase of 8.2% compared to the previous year [24]. However, meagre production is highly impacted by stressful conditions, due to the susceptibility of this species to stress and its impact on growth performance and the overall welfare of animals. For this reason, the use of nutritional tools to reduce meagre stress vulnerability must be studied and optimized. Recently, studies in meagre showed that Trp-enriched diets could mitigate stress responses [17,25].

Thus, this study aimed to assess the potential role of both a non-supplement diet and dietary Trp supplementation on growth performance, feed utilization, and metabolic responses in meagre (*Argyrosomus regius*) juveniles.

## 2. Materials and Methods

The experiment was conducted at the experimental facilities of Marine Zoological Station, University of Porto, Portugal. Trained scientists (following FELASA category C recommendations) directed the trial, and all procedures were conducted according to the recommendations of the European Union Directive (2010/63/EU) while operating under the Portuguese Law (DL 113/2013) on the protection of animals for scientific purposes. The experimental protocol was approved by the Animal Welfare and Ethics Body committee of the Interdisciplinary Centre of Marine and Environmental Research (ORBEA-CIIMAR, reference ORBEA_CIIMAR_27_2019).

### 2.1. Experimental Diets

Four experimental diets were formulated to be isoproteic (45%DM) and isolipidic (16%DM), with fish meal and fish oil as main protein and lipid sources, respectively. One diet was not supplemented with L-Trp (control diet), and three other diets were supplemented with graded levels of L-tryptophan (diets 0.25Trp, 0.5Trp and 1Trp, respectively), corresponding to a total dietary tryptophan of 1.06, 1.70, 2.08 and 3.24 g 16 $g^{-1}$ N, respectively. L-Tryptophan was coated with agar before being mixed with the other ingredients

to avoid leaching and delays in absorption in the digestive tract. Agar was dissolved in boiling distilled water and cooled to 40 °C before mixing. All ingredients were then finely ground, thoroughly mixed, and pelleted using a laboratory pellet mill (CPM: California Pellet, Mill, Crawfordsville, IN, USA) using a 2.5 mm die. Diets were dried in an oven at 35 °C for 24 h and stored at −20 °C until use. The formulation and proximate composition of the diets are presented in Table 1.

**Table 1.** Formulation and proximate analyses (% dry matter) of the experimental diets.

| Diets | 0Trp | 0.25Trp | 0.5Trp | 1Trp |
|---|---|---|---|---|
| *Ingredients (% DM)* | | | | |
| Fish meal [1] | 35.0 | 35.0 | 35.0 | 35.0 |
| Corn gluten [2] | 10.0 | 10.0 | 10.0 | 10.0 |
| Soybean meal [3] | 18.1 | 17.7 | 17.3 | 16.4 |
| Wheat meal [4] | 20.6 | 20.8 | 20.9 | 21.3 |
| Fish oil | 11.8 | 11.8 | 11.8 | 11.8 |
| Vitamin premix [5] | 1.0 | 1.0 | 1.0 | 1.0 |
| Choline chloride (50%) | 0.5 | 0.5 | 0.5 | 0.5 |
| Mineral premix [6] | 1.0 | 1.0 | 1.0 | 1.0 |
| Binder [7] | 1.0 | 1.0 | 1.0 | 1.0 |
| Agar | 1.0 | 1.0 | 1.0 | 1.0 |
| L-tryptophan | - | 0.25 | 0.50 | 1.00 |
| *Proximate analyses (% dry weight)* | | | | |
| Dry matter (%) | 96.6 | 95.8 | 95.7 | 95.7 |
| Crude protein | 46.3 | 45.8 | 45.7 | 45.9 |
| Crude lipid | 16.5 | 16.3 | 16.5 | 16.5 |
| Ash | 10.0 | 10.2 | 10.1 | 10.2 |

[1] Pesquera Centinela, Steam Dried LT, Chile (CP: 71.4%; CL 9.3%), Sorgal, S.A., Ovar, Portugal; [2] Corn gluten (CP: 72.2%; CL: 2.0%), Sorgal, S.A., Ovar, Portugal; [3] Soybean meal (CP: 54.9%; CL:2.1%), Sorgal, S.A., Ovar, Portugal; [4] Wheat gluten (CP: 13.9%; CL: 1.8%), Sorgal, S.A., Ovar, Portugal; [5] Vitamins (mg kg$^{-1}$ diet): retinol, 18,000 (IU kg$^{-1}$ diet); calciferol, 2000 (IU kg$^{-1}$ diet); alpha tocopherol, 35; menadion sodium bis., 10; thiamin, 15; riboflavin, 25; Ca pantothenate, 50; nicotinic acid, 200; pyridoxine, 5; folic acid, 10; cyanocobalamin, 0.02; biotin, 1.5; ascorbyl monophosphate, 50; inositol, 400.; [6] Minerals (mg kg$^{-1}$ diet): cobalt sulphate, 1.91; copper sulphate, 19.6; iron sulphate, 200; sodium fluoride, 2.21; potassium iodide, 0.78; magnesium oxide, 830; manganese oxide, 26; sodium selenite, 0.66; zinc oxide, 37.5; dicalcium phosphate, 8.02 (g kg$^{-1}$ diet); potassium chloride, 1.15 (g kg$^{-1}$ diet); sodium chloride, 0.4 (g kg$^{-1}$ diet); [7] Aquacube, Agil, UK.

### 2.2. Growth Trial

Meagre juveniles were obtained from IPMA's Aquaculture Research Station in Olhão and were kept in quarantine for two weeks. Afterward, the fish were transported to the experimental systems and adapted to the experimental conditions for two weeks. During both periods, the fish were fed a commercial diet (48% protein and 17% lipids, Sorgal, S.A., Ovar, Portugal).

The growth trial was conducted in a thermoregulated semi-recirculating water system, using 12 fiberglass cylindrical tanks of 100 L water capacity, supplied with a continuous flow of filtered seawater. Throughout the trial, water quality parameters were monitored; temperature, salinity, and oxygen levels were checked daily, and nitrogen compounds were measured three times a week. The temperature averaged $22.0 \pm 0.5$ °C, salinity averaged 35‰, and oxygen levels were kept near saturation.

At the beginning of the trial, fish had an average body weight of $36 \pm 2$ g. A total of 180 meagre were randomly distributed into 12 homogenous groups of 15 fish. The experimental diets were randomly assigned to triplicate groups of fish. The fish were fed to apparent visual satiation twice a day, six days a week, for 8 weeks. Utmost care was taken to avoid feed losses.

### 2.3. Sampling

At the beginning and at the end of the trial (56 days later), following one day of feed deprivation the fish were bulk-weighed after being anesthetized with ethylene glycol

monophenyl ether (0.3 mL L$^{-1}$). From the initial stock population, eight fish were collected. At the end of the trial, 3 fish from each tank were also euthanized with lethal doses of anesthetic (10 mL L$^{-1}$). These fish were then used for whole-body composition analysis and the weights of the whole fish, viscera, and liver were each recorded for hepatosomatic and visceral indices.

Five days after the end of the trial, a time course blood sample was performed. Two fish per tank (6 per dietary treatment) were randomly sampled at 0 h, 1 h, 3 h, 6 h, and 12 h after feeding. Following fish anesthesia, blood samples were taken from the caudal vein with heparinized syringes, and plasma was recovered using centrifugation (1000× $g$, 10 min) and kept at −20 °C to await further analysis. At the 6 h sampling point, fish were euthanized with a sharp blow to the head and immediately eviscerated in an ice-cold tray. The liver was excised, immediately frozen in liquid nitrogen, and then stored at −80 °C, until measurement of amino acid catabolism.

### 2.4. Analytical Methods

#### 2.4.1. Proximate Analysis

Chemical analysis of the ingredients, diets, and whole body composition was conducted in duplicate following standard methodology [26]: the dry matter obtained by drying the samples at 105 °C until a constant weight; protein content (N × 6.25) using the Kjeldahl method following acid digestion and distillation (Tecator System, Högamäs, Sweden; extraction unit model 1015 and 1026, respectively); lipid content extracted with petroleum ether using a Soxtec system (Tecator Systems, Höganäs, Sweden; extraction unit model 1043 and service until model 1046); ash acquired by means of incineration in a muffle furnace at 450 °C for 16 h.

The amino acid content of the experimental diets was analyzed in duplicate using high-performance liquid chromatography (HPLC) according to the Pico–Tag method. Samples were hydrolyzed for 23 h with 6N hydrochloric acid at 110 °C under an N$_2$ atmosphere, and derivatized with phenylisothiocyanate reagent (PITC, Ref. WAT088120, Waters™, Milford, MA, USA) before being separated using gradient exchange chromatography at 46 °C (Waters auto sample model 717 plus, Waters binary pump model 1525, Waters dual absorbance detector model 2487), according to the Pico–Tag method. Norleucine was used as an internal standard. Chromatographic peaks were identified, integrated, and quantified by comparing them to a known amino acid standard, using the Waters Breeze software package (Pierce NC10180, Waters™, Milford, MA, USA, 2001). Tryptophan was measured with a spectrophotometric method, as described by De Vries et al. [27]. Amino acid proximate composition is shown in Table 2.

#### 2.4.2. Plasma Metabolites

Commercial kits from Spinreact, S.A. (Gerona, Spain) were used to determine plasma cholesterol (ref. 1001090), glucose (ref. 1001190), and triglycerides (ref. 41031). Plasma protein was determined according to Bradford [28], using bovine serum albumin as standard.

#### 2.4.3. Enzyme Activity

Alanine aminotransferase (ALAT; EC 2.6.1.2), aspartate aminotransferase (ASAT; EC 2.6.1.1.), and glutamate dehydrogenase (GDH; EC 1.4.1.2.) activities were assessed in the liver. The livers were homogenized (dilution1/100) in an ice-cold buffer (30 mM HEPES, 0.25 mM saccharose, 0.5 mM EDTA, 5 mM K$_2$HPO$_4$, 1 mM dithiothreitol, pH 7.4). After 900× $g$ for 10 min centrifugation, the resultant supernatant was sonicated for 1 min (pulse 1 s, amplitude 50) and centrifuged again at 15,000× $g$ for 20 min. The resultant supernatant was separate for ALAT, ASAT, and GDH activity measurements. The GDH activity was measured using 10 mM of L-glutamic acid, as described previously by Bergmeyer (1974), at 37 °C. ALAT and ASAT were assayed with kits from Enzyline (ALAT/GPT, ref. 63313; ASAT/GOT, ref. 63213, bioMérieux, S.A., France) at 37 °C and followed at 340 nm. Enzyme activity was expressed as nmol of substrate transformed per min and per mg of protein

under the assay conditions. Soluble protein concentration was determined using the Bradford method [27], with bovine serum albumin used as a standard.

**Table 2.** Amino acid composition (g 16 g$^{-1}$ N) of the experimental diets.

| Diets | | 0Trp | 0.25Trp | 0.5Trp | 1Trp |
|---|---|---|---|---|---|
| Essential amino acids | | | | | |
| | Histidine | 3.05 | 3.09 | 3.04 | 2.99 |
| | Arginine | 6.09 | 6.10 | 6.12 | 6.14 |
| | Isoleucine | 4.32 | 4.43 | 4.50 | 4.33 |
| | Leucine | 8.83 | 8.82 | 8.91 | 8.99 |
| | Valine | 5.56 | 5.45 | 5.36 | 5.24 |
| | Lysine | 6.34 | 6.43 | 6.20 | 6.26 |
| | Methionine | 2.67 | 2.67 | 2.50 | 2.69 |
| | Phenylalanine | 4.87 | 4.74 | 4.85 | 4.61 |
| | Threonine | 4.32 | 4.27 | 4.24 | 4.25 |
| | Tryptophan | 1.06 | 1.70 | 2.08 | 3.24 |
| Nonessential amino acids | | | | | |
| | Aspartic Acid | 9.00 | 9.02 | 9.93 | 9.97 |
| | Glutamic Acid | 16.4 | 16.3 | 16.4 | 16.2 |
| | Serine | 5.56 | 5.43 | 5.70 | 4.75 |
| | Glycine | 5.57 | 5.41 | 5.57 | 5.09 |
| | Tyrosine | 3.88 | 3.93 | 3.75 | 3.73 |
| | Alanine | 6.13 | 6.52 | 6.21 | 5.72 |
| | Proline | 5.20 | 5.12 | 5.12 | 5.43 |

*2.5. Statistical Analysis*

Data were checked for normal distribution and homogeneity of variances, and normalized, if necessary. Statistical data analysis was conducted using a one-way analysis of variance (one-way ANOVA) at a probability level of $p \leq 0.05$. Significant differences among mean values were determined using the Tukey multiple comparison test. To analyze plasma metabolites, a two-way analysis of variance was used (two-way ANOVA), with diets and time as fixed factors. When the interaction was significant, diet and time effects were analyzed by performing a one-way ANOVA. All statistical analyses were performed using SPSS version 28.0 for Windows software package (IBM, Armonk, NY, USA, 2022).

**3. Results**

*3.1. Growth Trial*

The fish promptly accepted the experimental diets. Mortality was low and was not affected by dietary treatment (Table 3). Diet 1Trp promoted a lower weight gain, significantly affecting both the final body weight and the daily growth index compared to other experimental diets. Even though feed intake was similar in all diets, diet 1Trp also significantly reduced both feed efficiency and the protein efficiency ratio (Table 3). Regarding nitrogen (N) utilization, N intake decreased with increasing levels of Trp in the experimental diets, with the significantly lowest value in diet 1Trp. As for N retention (g kg ABW$^{-1}$ day$^{-1}$ and % N intake), these parameters decreased with increasing levels of Trp and were lower in fish fed the 1Trp diet. Feed intake linearly decreased with the increase in Trp dietary levels ($R^2$ = 0.6, $p$ = 0.005).

**Table 3.** Growth performance and feed utilization efficiency of meagre fed the experimental diets.

| Diets | 0Trp | 0.25Trp | 0.5Trp | 1Trp |
|---|---|---|---|---|
| Initial body weight (g) | $36.1 \pm 0.0$ | $36.1 \pm 0.0$ | $36.2 \pm 0.0$ | $36.0 \pm 0.0$ |
| Final body weight (g) | $123.7 \pm 7.0$ [b] | $116.5 \pm 1.8$ [b] | $114.0 \pm 2.9$ [b] | $87.8 \pm 3.0$ [a] |
| Weight gain (g kg $ABW^{-1\S}$ $day^{-1}$) | $19.5 \pm 0.7$ [b] | $18.8 \pm 0.2$ [b] | $18.5 \pm 0.3$ [b] | $14.9 \pm 0.5$ [a] |
| Thermal growth coefficient [1] | $0.14 \pm 0.0$ [b] | $0.13 \pm 0.0$ [b] | $0.13 \pm 0.0$ [b] | $0.09 \pm 0.0$ [a] |
| Daily growth index [2] | $3.0 \pm 0.2$ [b] | $2.8 \pm 0.1$ [b] | $2.8 \pm 0.1$ [b] | $2.0 \pm 0.1$ [a] |
| Feed intake (g kg $ABW^{-1\S}$ $day^{-1}$) | $17.5 \pm 0.4$ | $17.4 \pm 0.3$ | $16.8 \pm 0.2$ | $15.8 \pm 0.6$ |
| Feed efficiency [3] | $1.1 \pm 0.0$ [b] | $1.1 \pm 0.0$ [b] | $1.1 \pm 0.0$ [b] | $0.93 \pm 0.2$ [a] |
| Protein efficiency ratio [4] | $2.4 \pm 0.0$ [b] | $2.4 \pm 0.0$ [b] | $2.4 \pm 0.0$ [b] | $2.03 \pm 0.4$ [a] |
| Mortality (%) | $2.2 \pm 2.2$ | $0.0 \pm 0.0$ | $2.2 \pm 2.2$ | $4.4 \pm 2.2$ |
| *Nitrogen utilization* | | | | |
| N Intake (g kg $ABW^{-1}day^{-1}$) [5] | $1.3 \pm 0.1$ [b] | $1.3 \pm 0.0$ [ab] | $1.2 \pm 0.0$ [ab] | $1.2 \pm 0.1$ [a] |
| N Retention (g kg $ABW^{-1}day^{-1}$) [6] | $0.6 \pm 0.0$ [b] | $0.6 \pm 0.0$ [b] | $0.6 \pm 0.0$ [b] | $0.4 \pm 0.0$ [a] |
| N Retention (% nitrogen intake) [7] | $45.3 \pm 3.0$ [b] | $45.0 \pm 0.8$ [b] | $45.4 \pm 1.3$ [b] | $37.7 \pm 2.0$ [a] |

Mean values $\pm$ SD (n = 3) in the same row with different superscript letters are significantly different ($p < 0.05$). An absence of different superscript letters indicates no significant differences ($p \geq 0.05$). [§] Average body weight (ABW): initial body weight (IBW) + final body weight (FBW)/2. [1] Thermal growth coefficient = (($FBW^{1/3}$ − $IBW^{1/3}$)/(time in days × °C)) × 100. [2] DGI: (($FBW^{1/3}$ − $IBW^{1/3}$)/time in days) × 100. [3] FE: wet weight gain/dry feed intake. [4] PER: wet weight gain/dry crude protein intake. [5] Nitrogen intake (g kg $ABW^{-1}$ $day^{-1}$) = nitrogen/ABW × time in days. [6] Daily nitrogen retention (g Kg $ABW^{-1}$ $day^{-1}$) = (FBW × Final body nitrogen (%)) − (IBW × Initial body nitrogen)/(((IBW + FBW)/2) × time in days. [7] Nitrogen retention (% nitrogen intake) = (nitrogen retention/nitrogen intake) × 100.

Whole-body protein and ash content were unaffected by different dietary Trp levels. However, whole-body lipids and dry matter were higher in the control than in the 1Trp diet, both decreasing with increasing Trp levels. Both the hepatosomatic and visceral indices and the relative intestine length were unaffected by dietary treatments (Table 4).

**Table 4.** Whole-body composition, hepatosomatic index and visceral index of meagre fed the experimental diets.

| Diets | Initial | 0Trp | 0.25Trp | 0.5Trp | 1Trp |
|---|---|---|---|---|---|
| *Whole-body composition (%)* | | | | | |
| Dry matter (%) | 24.6 | $28.5 \pm 0.3$ [b] | $27.81 \pm 1.0$ [ab] | $28.3 \pm 0.4$ [ab] | $26.4 \pm 1.0$ [a] |
| Protein | 15.8 | $17.5 \pm 0.4$ | $17.9 \pm 0.2$ | $18.0 \pm 0.4$ | $17.1 \pm 0.3$ |
| Lipids | 5.9 | $8.1 \pm 0.3$ [b] | $7.2 \pm 1.2$ [ab] | $7.8 \pm 0.3$ [ab] | $6.1 \pm 0.8$ [a] |
| Ash | 4.2 | $4.3 \pm 0.4$ | $4.1 \pm 0.3$ | $4.4 \pm 0.2$ | $4.3 \pm 0.1$ |
| *Body indices* | | | | | |
| Hepatosomatic index [1] | — | $1.7 \pm 0.3$ | $1.8 \pm 0.3$ | $2.0 \pm 0.1$ | $1.9 \pm 0.4$ |
| Visceral index [2] | — | $4.0 \pm 0.4$ | $3.9 \pm 0.4$ | $4.1 \pm 0.2$ | $4.1 \pm 0.6$ |
| Relative intestinal length [3] | — | $0.2 \pm 0.1$ | $0.2 \pm 0.1$ | $0.1 \pm 0.0$ | $0.2 \pm 0.1$ |

Mean values $\pm$ SD (n = 3) in the same row with different superscript letters are significantly different ($p < 0.05$). An absence of different superscript letters indicates no significant differences ($p \geq 0.05$). [1] HSI: (liver weight/body weight) × 100. [2] VI: (viscera weight/body weight) × 100. [3] RIL = intestine length (mm)/body weight (g).

### 3.2. Plasma Metabolites

Concentrations of plasma metabolites are presented in Table 5. All metabolites, except for plasma glucose, followed the same trend over time, irrespective of dietary treatment. Plasma protein levels peaked 1 h after feeding, remaining high until 12 h after feeding. Plasma cholesterol also peaked at 1 h after feeding but reached a basal level at 12 h. Plasma triglycerides showed an interaction between diet and time (Table 5), with a peak 1 h post-feeding irrespective of dietary treatment and then decreasing significantly until 12 h after feeding.

**Table 5.** Time-course effect of the experimental diets on plasma metabolites concentrations (mg /dL) of meagre.

| Diets | Time | | | | |
|---|---|---|---|---|---|
| | 0h | 1h | 3h | 6h | 12h |
| **Protein** | | | | | |
| 0Trp | 1.8 ± 0.2 | 2.3 ± 0.2 | 2.3 ± 0.1 | 2.4 ± 0.2 | 2.6 ± 0.4 |
| 0.25Trp | 1.8 ± 0.2 | 2.3 ± 0.2 | 2.4 ± 0.2 | 2.4 ± 0.1 | 2.3 ± 0.1 |
| 0.5Trp | 1.8 ± 0.1 | 2.2 ± 0.2 | 2.5 ± 0.2 | 2.3 ± 0.1 | 2.3 ± 0.2 |
| 1Trp | 1.8 ± 0.2 | 2.3 ± 0.2 | 2.0 ± 0.4 | 2.2 ± 0.1 | 2.3 ± 0.3 |
| **Glucose** | | | | | |
| 0Trp | 71.2 ± 23.3 [AB] | 44.0 ± 9.1 [A] | 69.1 ± 11.3 [bAB] | 72.5 ± 23.8 [B] | 112.4 ± 26.2 [bC] |
| 0.25Trp | 77.0 ± 4.8 [B] | 48.1 ± 9.6 [A] | 48.6 ± 17.3 [aA] | 69.9 ± 8.2 [AB] | 63.7 ± 10.8 [aAB] |
| 0.5Trp | 76.4 ± 7.7 | 45.0 ± 11.4 | 49.9 ± 7.8 [a] | 75.9 ± 16.5 | 63.1 ± 18.4 [a] |
| 1Trp | 60.5 ± 3.7 | 46.5 ± 6.7 | 55.5 ± 21.4 [ab] | 72.7 ± 16.4 | 58.7 ± 15.2 [a] |
| **Cholesterol** | | | | | |
| 0Trp | 67.3 ± 8.0 | 109.8 ± 14.3 | 73.3 ± 5.1 | 78.8 ± 7.5 | 71.3 ± 6.5 |
| 0.25Trp | 64.1 ± 4.8 | 112.2 ± 10.5 | 87.0 ± 9.9 | 88.8 ± 10.2 | 63.1 ± 5.5 |
| 0.5Trp | 67.1 ± 7.7 | 112.1 ± 20.5 | 89.9 ± 14.4 | 87.6 ± 12.4 | 66.1 ± 6.1 |
| 1Trp | 63.1 ± 3.7 | 100.0 ± 19.7 | 80.0 ± 13.4 | 75.6 ± 9.4 | 65.2 ± 10.8 |
| **Triglycerides** | | | | | |
| 0Trp | 153.6 ± 27.0 [A] | 735.8 ± 84.4 [C] | 486.8 ± 66.6 [B] | 575.1 ± 103.1 [B] | 541.8 ± 92.6 [B] |
| 0.25Trp | 152.5 ± 15.1 [A] | 607.3 ± 46.6 [C] | 387.6 ± 48.0 [B] | 720.1 ± 92.1 [C] | 534.4 ± 171.7 [BC] |
| 0.5Trp | 161.8 ± 20.9 [A] | 819.5 ± 152.0 [D] | 637.5 ± 60.2 [CD] | 595.4 ± 80.0 [BC] | 274.0 ± 102.7 [AB] |
| 1Trp | 137.3 ± 8.2 [A] | 689.5 ± 95.5 [D] | 435.4 ± 89.4 [C] | 636.4 ± 70.3 [D] | 279.7 ± 66.7 [B] |

| Two Way ANOVA | Variation Source | | | Diet | | | | Time | | | | |
|---|---|---|---|---|---|---|---|---|---|---|---|---|
| | Time | Diet | Interaction | 0Trp | 0.25Trp | 0.5Trp | 1Trp | 0h | 1h | 3h | 6h | 12h |
| Protein | *** | ns | ns | a | a | a | a | a | b | b | b | b |
| Glucose | *** | ** | ** | | | | | | | | | |
| Cholesterol | *** | * | ns | ab | ab | b | a | a | c | b | c | a |
| Triglycerides | *** | ns | *** | | | | | | | | | |

Mean values ± SD (n = 6). Two-way ANOVA: ns: non-significant differences ($p \geq 0.05$); * $p < 0.05$; ** $p < 0.01$; *** $p < 0.001$. If interaction was significant, diet and time effects were analyzed using a one-way ANOVA; mean values in the same column with different lowercase letters indicate significant differences among diets ($p < 0.05$) for each time; within the same row, mean values with different capital letters indicate significant differences between time ($p < 0.05$) for each diet.

Irrespective of the sampling time, the dietary Trp levels did not affect plasma protein and triglycerides. However, the plasma cholesterol levels of fish fed the 0.5Trp diet were significantly higher than those of fish fed the 1Trp diet.

Plasma glucose showed an interaction between diet and time. Glucose levels significantly decreased 1 h after feeding for the control and 0.25Trp diets, increasing afterward. The glucose plasma levels in the control diet and 0.25Trp diets peaked at 12 h and 6 h after feeding, respectively, then decreased below basal levels. At 3 h after feeding, glucose levels of the 0.25Trp and 0.5Trp diets were lower than that of the control diet. Similarly, at 12 h after feeding, dietary treatment also affected glucose levels, with the control diet having significantly higher glucose levels than the remaining three supplemented diets.

### 3.3. Hepatic Protein Catabolism Enzyme Activity

Dietary inclusion of Trp affected the enzymes measured (Table 6). All enzymes (ALAT, ASAT, and GDH) showed a similar pattern in response to the inclusion of Trp in diets. The highest activity was observed in the control diet (0Trp), with decreasing activities for all enzymes as the level of dietary tryptophan inclusion increased. Consequently, the lowest level of activity was observed in the most highly supplemented diet; namely, diet 1Trp.

**Table 6.** Hepatic activity of aspartate aminotransferase (ASAT), alanine aminotransferase (ALAT), and glutamate dehydrogenase (GDH) enzymes of meagre juveniles fed the experimental diets.

| Diets | 0Trp | 0.25Trp | 0.5Trp | 1Trp |
|---|---|---|---|---|
| ASAT | 1947.6 ± 198.6 [c] | 1673.1 ± 104.4 [bc] | 1529.4 ± 211.7 [b] | 1231.0 ± 132.7 [a] |
| ALAT | 666.0 ± 83.4 [c] | 582.8 ± 38.3 [bc] | 534.7 ± 88.5 [b] | 416.0 ± 24.1 [a] |
| GDH | 830.6 ± 71.6 [c] | 721.3 ± 53.3 [b] | 684.2 ± 51.6 [ab] | 592.3 ± 53.6 [a] |

Mean values ± SD (n = 6) in the same row with different superscript letters are significantly different ($p < 0.05$). Units for all activities are presented as nmol/min/mg protein.

## 4. Discussion

Many practices deemed necessary to aquaculture unavoidably submit fish to several stressors. In response to a stressor, fish undergo several biochemical and physiological changes to compensate for the imposed challenge, which can impair the fish's nutritional and health status [29]. Tryptophan (Trp) has been pointed out as a possible nutritional strategy to mitigate stress through use as a feed additive, due to being the precursor of serotonin and melatonin [10,30–34]. Previous studies in meagre showed that dietary supplementation with Trp (total dietary level of 0.11% diet) modulated their immune parameters following acute stress [3]. Moreover, dietary supplementation with 0.25% L-Trp did not affect meagre growth performance and enhanced their observed mucus antibacterial activity [18] Correspondingly, this study evaluated the effect of a higher Trp dietary supplementation (0.25, 0.5 and 1% of the diet) on meagre performance and feed utilization, as well as on the plasma metabolites profile and the key liver enzymes involved in amino acid catabolism.

Tryptophan requirements differ for each fish species and have been reported to range from 0.3 to 1.3% of dietary protein [34,35]. Moreover, these requirements may increase with stress [34]. To our knowledge, the nutritional requirements of Trp for meagre have yet to be established. In some species, an excess of dietary Trp (i.e., above their requirement levels) may increase serotonin and melatonin production [30,32], which can lead to a decrease in voluntary feed intake [36,37]. Murthy and Vanghese [38] determined that for *Labeo rohita* the optimum Trp level was 1.13%, and fish fed a diet comprising 1.38% Trp presented a decreased level of growth; the authors attributed this to a possible toxic effect, which will be further discussed. Hseu et al. [8] also showed that supplementation of Trp altered feeding behavior, with a decrease in feed intake observed with supplementation level of Trp higher than 0.5% of the diet. Recent studies have also suggested that growth performance may be reduced by higher dietary Trp levels [39–41]. Herrera et al. [18] showed that supplementation up to 0.25% of the total diet did not affect growth performance nor feed efficiency in meagre. In this study, dietary supplementation levels of Trp up to 0.5% (diet 0.5Trp; 2.08 g 16 $g^{-1}$ N total dietary Trp level) did not affect growth performance and feed utilization. However, 1% Trp supplementation (diet 1Trp; 3.24 g 16 $g^{-1}$ N total dietary Trp level) decreased both these parameters, suggesting that a total Trp dietary level of 3.24 g 16 $g^{-1}$ N may be harmful to this species. As aforementioned, high levels of some amino acids could be considered toxic. This may be due to excessive levels exerting antagonistic actions against other amino acids, i.e., may impair the absorption and utilization of other amino acids, resulting in a deficient absorption rate of this amino acid. Trp is a large neutral amino acid (LNAA), competing for the same transporter as other LNAAs, such as valine, leucine, isoleucine, tyrosine, and phenylalanine [42]. Supplementation of Trp can affect the Trp/LNAA ratio, meaning that Trp brain concentrations may increase at the expense of other LNAAs [20,32,43,44].

The whole-body lipid composition of meagre decreased in the fish group feeding on 1Trp, while whole-body protein levels were maintained regardless of dietary treatment. The supplied diets from diet 0Trp to 0.5Trp were enough both for energy expenditure and lipid deposition. However, fish fed the 1Trp diet had lower ingestion, so less energy was available for energetic metabolism, decreasing the amount allocated to lipid deposition.

Similar results were reported for rats, with high supplementation of Trp reducing fat accumulation in the body, which could be to higher oxidation of fatty acids [45].

In addition to measuring cortisol and catecholamine levels, secondary stress response parameters (such as plasma glucose levels) have been used [46–48] to evaluate the stress response. Under stress, the release of cortisol, catecholamines, and glucagon hormones increases the metabolic state, and glucose functions as the main energy source to cope with this increased energy demand [49]. In fish, increased plasma glucose levels are often associated with increased plasma cortisol levels [50]. This relationship between Trp and cortisol regulation has been attributed to the interaction between serotonergic activity and the hypothalamic–pituitary–interrenal axis. Therefore, glucose levels are expected to increase upon stress [49]. In fact, reports of Trp-enriched diets fed to fish showed that stress markers such as cortisol and glucose showed similar tendencies, with increased levels of Trp in the diets ultimately decreasing both serum cortisol and blood glucose in data for diets supplemented with Trp [31,51]. In the present study, the supplementation of Trp lowered glucose levels 3 h after feeding compared to 0 h after feeding. These results evidence a probable reduction of the stress condition of fish that were fed the Trp-supplemented diets, reflecting a reduction in body energy mobilization. Similarly, feeding *Cyprinus carpio* a diet supplemented with 5% Trp has been shown to result in lower cortisol and glucose levels compared to a control group after copper exposure, yet levels of cortisol and glucose did not reach pre-exposure levels [9]. In *Cirrhinus mrigala*, plasma cortisol and blood levels were affected by dietary treatment up to 2.72% DM Trp supplementation, in a similar observable trend; the supplementation of L-Trp gradually decreasing both cortisol and glucose levels [31]. In rohu, cortisol and glucose levels decreased with the increase of dietary Trp level from 0% up to 1.42% of their total diet [51]. In gilthead seabream, supplemented diets of 1.5 and 2% DM (total dietary Trp of 0.54 and 0.78% DM) decreased glucose levels in line with increasing Trp levels, a result observed in both stressed and unstressed fish [52]. In rainbow trout exposed to stress, the Trp supplemented diet (at 3.57% of the total dietary level) reduced cortisol levels compared to the control diet. Interestingly, in the same study, fish not submitted to stress showed higher cortisol levels when fed Trp supplemented diets [17]. The authors attributed this evaluation to other underlying mechanisms involved in mediating the roles of Trp on stress response [16]. In addition, the rainbow trout showed a decrease in post-stress plasma cortisol when fed a Trp supplemented diet (total dietary Trp level of 3.57%) for 7 days [53]. In Atlantic salmon, increasing dietary Trp levels by 3-fold (1.2% Trp) resulted in decreased plasma cortisol levels compared with those fed a 2-fold and one-fold increase [54].

Stress conditions induce a hypermetabolic status which may induce the mobilization of amino acids as a coping mechanism for the increased energy demand. Normally, increased activity of GDH, ASAT, and ALAT are used as indicators of increased catabolic utilization of dietary protein by fish, as the amino acids which are not used for protein synthesis are deaminated [55]. In the present study, hepatic amino acid catabolism enzyme activity was affected by dietary treatment, decreasing with the increased level of Trp inclusion in the diets. These results are in line with those reported in meagre by Herrera et al. [17], who observed a decreased activity of amino acid catabolic enzymes with the augmentation of Trp levels. However, other authors have reported the opposite. For example, in Atlantic cod (*Gadus morhua*) and Senegalese sole (*Solea senegalensis*), it has been shown that the dietary supplementation of Trp increased the activity of these enzymes [56,57], as the dietary excess of amino acids induced the activity of protein-metabolizing enzymes, particularly GDH and ALAT [58].

As aforementioned, nitrogen retention was lower in fish fed diets of 1% Trp, while the level of protein catabolism activity decreased. Thus, it can be hypothesized that the reduction of nitrogen retention in fish fed the 1Trp diet was not due to increased use of protein for energetic purposes but rather to a deficient protein intake associated with a reduced feed intake. Our results correlate well with the reduction of amino acid catabolism enzymes, which may indicate a reduction in the meagres' metabolic rate, reducing energy

expenditure and with it the use of amino acids for energetic purposes. The discrepancies between these results and those of the different studies found in the literature could be due to several aspects, such as species, the nutritional status of the animals, levels of supplementation, developmental stage, and stress condition.

### 5. Conclusions

Overall, our results suggested that a dietary Trp-level increase of up to 2.08 g 16 $g^{-1}$ N did not affect growth performance and feed efficiency, but that these parameters were severely compromised with 3.24 g 16 $g^{-1}$ N Trp level. Voluntary feed intake linearly decreased with the increase in dietary Trp level, suggesting a modulatory effect of feeding behavior. Moreover, to evaluate secondary stress responses, glucose values were determined; fish fed Trp diets showed lower levels of glucose than those fed the control diet. Thus, it can be recommended that up to 2.08 g 16 $g^{-1}$ N can be supplemented in diets for meagre without compromising performance and feed utilization.

Further studies would be needed to clarify the requirements of this amino acid for this species, and to understand the underlying mechanisms of Trp utilization by this species.

**Author Contributions:** A.P.-J., H.P. and A.O.-T. conceived the study and designed the trial; R.P., C.C. and R.M. carried out the experiments; C.T., R.P. and C.C. carried out plasma and enzyme analysis; C.T., H.P. and A.P.-J. analyzed the data; C.T. and A.P.-J. wrote the manuscript, and all authors revised the manuscript; E.M. helped with the funding of the project. All authors have read and agreed to the published version of the manuscript.

**Funding:** This work was financed by the Ocean3R project (NORT-01-0145-FEDER-000064), supported by the North Portugal Regional Operational Program (NORT2020) under the PORTUGAL 2020 Partnership Agreement, and through the European Regional Development Fund (ERDF). The first author is supported by a PhD grant funded by the Foundation for Science and Technology (FCT) with the reference 2022.10117.BD.

**Institutional Review Board Statement:** The experimental protocol was approved by the Animal Welfare and Ethics Body committee of the Interdisciplinary Centre of Marine and Environmental Research (ORBEA-CIIMAR, reference ORBEA_CIIMAR_27_2019), in compliance with the European Union directive 2010/63/EU and the Portuguese Law (DL 113/2013).

**Data Availability Statement:** Not applicable.

**Conflicts of Interest:** The authors declare no conflict of interest.

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
