# Peer review of "Dietary Tryptophan Supplementation Implications on Performance, Plasma Metabolites, and Amino Acid Catabolism Enzymes in Meagre (Argyrosomus regius)"

_fishes, doi:10.3390/fishes8030141_

Round 1

Reviewer 1 Report

1. Abstract: What conclusions can be drawn from these results? The authors should provide the main conclusions in the abstract.

2. L77 According to the section of sampling (2.3), blood samples were collected at 0h, 1h, 3h, 6h, and 12h after feeding. So it should a 12h time course blood sampling.

3. L132 Please provide the concentration of ethylene glycol monophenyl ether for anesthesia.

4. L134 Please provide the concentration of lethal dose of anesthetic.

5. L137-139 How many fish were sampled at each sampling point?

6. L211 In table 3, please provide what formula are those parameters of nitrogen utilization calculated on?

7. L271 “from 0.3-1.3% of dietary protein”. Please provide references?

8. L303 Why didn't the authors measure cortisol level? It may be more reflective of the stress response.

9. L344 “... nitrogen retention, as well as nitrogen retention were lower...”?

10. L364-367 Feed intake decreased with the increasing of dietary Trp level. If linear regression analysis is performed, it may be possible to observe a negative effect of dietary tryptophan level on feeding rate. Thus it cannot be concluded that there is no compromise in feed utilization.

Reviewer 2 Report

Report on evaluation of  224 3934 ms

Abstract

-Make line 20 clearer ( add “but” to read ---- diet but supplemented with---

-Line 19 and 20 should read” graded levels of tryptophan, 0.49, 0.78, 0.95 and 1.49% DM, corresponding to control, 0.5Trp, 1Trp and 2Trp. If the doses given here were the ones used in the project, it is not clear where the values 2.49 and 3.24 %DM came from.

-Line25: add doses to read—although doses up to 1Tpr do not---

Lines 27 make the sentence clear

Introduction

-Line 34: replace routine farming with routine handling

-Line 37: delete the word “consequences”

-Make lines 37/38 clear to the reader

-Line 38 looks like it is not complete, especially the phrase “eventual parasites and infectious diseases”

Make lines 57 and 58 clear

-Line 64: Use feed conversion efficiency

-Line 68: compared to previous year

-Line 69 is not clear to the reader

-Line 77: Was the experiment contacted for 12 or 24h?

Materials and methods

-Line 91: Replace “and” with “while”

-Line 95: Add “in” to read, leaching and delay in absorption. Again is the final dose 1.49 or 2.49 or 3.24%DM?

-It should be made clear that sampling after 5 days was an extra experiment

-Line 137: Five days after the end of the first trial a second experiment was conducted to----

-Line 138: Anaesthized  fish cannot be experimented with!

-Line 150: Delete the word “units”

-Line 157 is not complete especially the phrase “and then pre-column

-It would be understandable to the reader if lines 157-160 were made more clearer to the reader

-Amino acid Table can be moved to the results section

Statistical analysis

-Use P≤0.05 instead of P<0.05, so that 0.05 is inclusive

-Line 191: Dose vs time response curves is a more suitable analytical tool than two-way anova

-Line 193: One way ANOVA should be used when there is no interaction but not the other way round

Results

-Lines 199- 200: Wight gain is repeated

Lines 199-200: It is not indicated whether the results were significant. Since values will always differ, it is important to indicate whether the differences are significant.

Thermal growth coefficient should have been used instead of Daily coefficient because it takes care of differences in temperature

-Indicate significance for feed intake and mortality

Table 4. What does whole-body mean? Clarity is very important

-Indicate significance for protein, ash and hepatosomatic index within the rows

-Initial values would have been very useful to set background information

Line 208: Use relative in place of relatively

--Line232: There is no clarity with what is called interaction. Line graphs can represent the information clearly. The variation of levels of the doses with time can be a better representation for each of the traits e.g protein, glucose etc. Confidence limits can be set to ascertain the differences between doses

-Line 234: It is not clear what aspect peaked in 12 and 6h

-Line 238: Two similar words ” than” and “compared” follow each other, delete one of them

Table 5. Insert the letters indicating statistical significance. While fixing the letters of significance, consistency is important. Use lower case but not something like bAB. What is the interpretation of those values without letters?

The meaning of the letters in Table 5 is not clear to the reader. There are no values in the ANOVA Table and it is not clear what the letters presented in this Table mean

-Line 252 is not clear

Table 6 is not well organized. It does not distinguish between enzymes and diets. The meaning of the terms ASAT, ALAT and  GDH should be stated

Table 6.

Enzyme

Dietary treatment

ASAT

Control

0.5Trp

1Trp

2Trp

ALAT

GHD

Discussion

-Line 261: Insert the word “changes” to read biochemical and physiological changes

-Lines 265 and 266: If 25% implies 1.61mg/g of feed. What does 1% stand for? Clarity and consistency is very important in communication

-Line 266-269: Do the authors aim to or they have already evaluated?

-Lines 270-294 appear more of a review than discussion on results of the present study

-Line 283: What does the term  zootechnical parameters mean?

-Line 286: Feed efficiency is usually less than 1. One cannot get more kg of fish than feed input!

-Line 286: It is not clear which was the highest levelof 2Trp, 2.49% or 3.24%? Make lines 286 and 287 clear

The argument offered in lines 295-300 does not make much appeal to scientific reality. If energy intake is reduced by lowered feed intake, does this not mean that body reserves will be mobilized for energy supply and the first body reserve to be mobilized is lipid?

-Line 300: Use and instead of therefore

-302: What was attributed to oxidation of fatty acids?

-Lines 314 and  315: Are not supported by Table 5

-Line 333 is not clear

-Line 334: Rephrase “fish that were fed----

Lines 340 and 341 contradict each other

-Line 344: Nitrogen retention is repeated

-Line 346: Make the sentence clear

-Line 347 does not make sense

-Line 350: Rephrase to read “increased energy demand”

-Line 356: Use “who” instead of “which”

-Line 357: Was plasma reduced or something in the plasma was reduced? Clarify

-Line 358: Is it true that if metabolic rate and energy expenditure are reduced amino acids are also reduced?

-Line 365: Was the inclusion level up to 2.49 or 3.24%? Clarify

_Line 366: “suggesting feed was not compromised” What does this phrase mean?

What is the recommended level of Trp in the present study?

Reviewer 3 Report

Review report fishes-2243934

Title:   Effects of dietary tryptophan supplementation in meagre (Argyrosomus regius) juveniles

Title: it needs modification since the theme is very informative so I strongly suggest the title as per the content but present title is not acceptable.

Abstract: It is not et al impressive in term of language and phrases so it needs to be revised. Line 16-17 it did not reflect the novelty and we all aware about the function of tryptophan so what is the novelty of this work need to be mentioned.

Introduction

The 80 percentage refences should be beyond 2018 onwards. So, I suggest to replace the older references with newer one.

Line no.76-77 are not necessary here as it is part of material and method so revise the last para of introduction. It should end with objective of the present study.

Material and methods

2.1. Experimental diets

Why only four diets statistically there  should be minimum 5 and how the level of inclusion of tryptophan was decided nothing is clear so need to mention in brief.

2.3. Sample collection

For how many days the experiment was performed and what was the experimental design was followed.

Results

Suggested to calculate the some more of the index like IEAAE or EAAI

References

Should be as per the journal format and in present form does not indicative journal formatting.

Round 2

Reviewer 1 Report

No.

Reviewer 3 Report

Now author meticulously has addressed all the comments what I have raised so now article in proper shape. Now it is acceptable in the Journal.